# Augmenting Social Science Research with Multimodal Data Collection: The EZ-MMLA Toolkit

**DOI:** 10.3390/s22020568

**Published:** 2022-01-12

**Authors:** Bertrand Schneider, Javaria Hassan, Gahyun Sung

**Affiliations:** Harvard Graduate School of Education, Harvard University, Cambridge, MA 02138, USA; javariahassan@gse.harvard.edu (J.H.); gsung@g.harvard.edu (G.S.)

**Keywords:** sensor applications and deployments, data mining, computer vision

## Abstract

While the majority of social scientists still rely on traditional research instruments (e.g., surveys, self-reports, qualitative observations), multimodal sensing is becoming an emerging methodology for capturing human behaviors. Sensing technology has the potential to complement and enrich traditional measures by providing high frequency data on people’s behavior, cognition and affects. However, there is currently no easy-to-use toolkit for recording multimodal data streams. Existing methodologies rely on the use of physical sensors and custom-written code for accessing sensor data. In this paper, we present the EZ-MMLA toolkit. This toolkit was implemented as a website and provides easy access to multimodal data collection algorithms. One can collect a variety of data modalities: data on users’ attention (eye-tracking), physiological states (heart rate), body posture (skeletal data), gestures (from hand motion), emotions (from facial expressions and speech) and lower-level computer vision algorithms (e.g., fiducial/color tracking). This toolkit can run from any browser and does not require dedicated hardware or programming experience. We compare this toolkit with traditional methods and describe a case study where the EZ-MMLA toolkit was used by aspiring educational researchers in a classroom context. We conclude by discussing future work and other applications of this toolkit, potential limitations and implications.

## 1. Introduction

Most fields of research heavily invest in the development of better scientific instruments for collecting fine-grained, reliable and accurate data. In Physics, the Large Hadron Colider is the world’s largest and most powerful particle accelerator; it took decades to build and cost more than 7.5 billion EUR [1]. In Astronomy, the James Webb Space Telescope is capable of observing the most distant events in the universe, and costs around 10 billion dollars [2]. In the social sciences, however, instruments for capturing human behaviors have not changed very much over the last century (e.g., questionnaires, interviews, human coding of video/audio data). While these instruments provide useful data for capturing social constructs, they are known to be subject to various cognitive and social biases (e.g., confirmation, selection, ordering, recency, priming, social desirability biases-to name a few).

An innovative way of complementing existing instruments in the social sciences is through multimodal sensing technologies. In education, for example, Multimodal Learning Analytics (MMLA, [3]) is starting to be used as a methodology for tracking learners’ states. MMLA research focuses on using high frequency sensors, such as eye-trackers and motion sensors, harnessing their ability to capture rich physiological and behavioral data [4]. Previously in this journal, Schneider et al. [5] have identified over 23 sensor devices that can be used in the domain of learning. Types of data that can be collected include body movements, gaze orientation, facial expressions and physiological data such as heart rate. However, high-quality sensor devices are often expensive and require domain expertise to understand and use [6]. Processing the generated data also requires skill and domain knowledge. In contrast, less-sophisticated modalities such as video and audio, which have a smaller cost and overhead in setting up, are limited by the quality of information they provide.

Considering these deficiencies, this paper presents the design, development and ecological use of a data collection website called the EZ-MMLA Toolkit. The website provides an intuitive, unintrusive, low-cost and ethical approach to collecting multimodal data using machine learning models that extract information from video and audio recordings. Currently, the modalities that can be collected through the website include body posture, facial landmarks, gaze location, emotion, heart rate and hand gestures.

In developing our toolkit, we accounted for important design and ethical considerations. First, the complexities of the machine learning models are hidden behind a user-friendly web interface that anyone can use irrespective of their level of technical knowledge. Moreover, the model computations happen directly in the browser, which means that sensitive video and audio data are not shared through a network and data privacy is maintained by the user. Lastly, our application allows for data collection through any browser and webcam, rather than through traditionally used expensive, individual physical sensors. This significantly reduces costs and makes multimodal data collection relatively inexpensive and broadly accessible to a wide range of users worldwide.

We further offer potential applications of our toolkit and present a case study where the website was used to teach a course on multimodal learning analytics. We report students’ feedback on the toolkit, describe the projects they worked on using the toolkit, and present preliminary research questions that can be explored by collecting and analyzing students’ multimodal data.

We first provide an overview of past literature in Section 2. We then describe conventional approaches to multimodal data collection in Section 3. The system description, case study, and preliminary research questions are explained in Section 4. Finally, we describe limitations and directions for future research in Section 5 and conclude the paper in Section 6.

In sum, the current paper contributes to the scientific community by introducing an intuitive, unintrusive, low-cost, and ethical approach to collecting multimodal data using cutting-edge machine learning models. Our implementation of this approach, the EZ-MMLA toolkit, differentiates itself from preexisting attempts to assist multimodal data collection by its accessible and open-source nature; prior software for multimodal data collection has largely been proprietary or designed to be used exclusively by programmers (Section 2.2). Our case study also demonstrates the toolkit’s potential to democratize sensor-data based research (Section 4), making multimodal data collection accessible to novice researchers and research groups lacking extensive resources (see Section 4.2 for a comparison with other toolkits).

## 2. Literature Review

### 2.1. Why Multimodal Sensing? Examples from Educational Research

Recently, data collection and analysis of multimodal data has become accessible and popular among social scientists. In educational research, the field of Multimodal Learning Analytics (MMLA) has helped researchers understand and model complex learning processes generally overlooked by traditional approaches. In this section, we provide examples showing how MMLA has been used to support research, teaching, and augment traditional learning environments.

MMLA has helped educational researchers reach new insights when analyzing multimodal data collected from learners. Motion data of students engaged in classroom activities can significantly predict learning outcomes [7]; linguistic, textual and speech data can provide useful information about students’ level of expertise [8]); and affective data has been correlated with students’ performance on standardized tests [9]. From a practical perspective, these insights can be used by educators to inform their teaching practices and personalize instruction. For example, gaze data can be used to enhance Interactive Tutoring Systems (ITS [10]). MMLA also has the potential to make statistics and data mining more accessible to non-technical students. Schneider, Reilly, & Radu [11] used multimodal sensors to teach MMLA concepts to educators and practitioners. Their findings suggest that multimodal data can create more engaging, relevant, hands-on learning activities compared to traditional data analysis courses.

Educational researchers have also used MMLA-based applications to support classroom learning and teaching. Maldonado et al. [12] used motion (i.e., Pozyx) sensors to track instructors’ location in lab settings. Teachers reported that the data was helpful for reflecting on their day-to-day work and for sharing best practices with others. Ref. [13] developed the Sensei system, which tracks children’s location in Montessori classrooms. The authors found that Sensei has the potential to augment teachers’ observations, help them plan personalized curricula for students, and identify children who receive less attention. A similar example for adult education is the EduSense platform [14] which generates behavioral data on students (e.g., body postures, speech time, head orientation). The data is available to teachers and can provide useful insights about students’ engagement and attention.

### 2.2. Current Toolkits Used to Conduct Multimodal Research

While this work is promising, multimodal sensing is not a widely used methodology in the social sciences because of the knowledge required to collect, process and analyze multimodal datasets. There are several significant obstacles that researchers need to overcome: (1) *cost*: while some sensors are relatively low cost, most are prohibitively expensive (e.g., a professional eye-tracker can cost thousands of dollars); (2) *data extraction*: either researchers use proprietary softwares (which can be more expensive than the sensor itself) or they create a software that can access the data via an API or other interface (which requires an engineer or expert coding skills); (3) *data synchronization*: oftentimes sensors are built to be used alone, and not in conjunction with other sensors (again, engineering skills are required to “fuse” different datasets together).

There are very few toolkits or software that facilitate multimodal data collection. iMotions [15], for example, is a proprietary software that provides an all-in-one solution for conducting multimodal research—but at a price that most researchers cannot afford with each modality costing a recurring subscription fee. The Social Signal Interpretation (SSI [16]) framework is an open-source project that offers tools to record, analyze and recognize human behavior in real-time. However, the framework is designed to be used by programmers and involves a steep learning curve (i.e., users need to go through a lengthy documentation to create simple data collection pipelines). Both toolkits require dedicated physical sensors (e.g., eye-trackers, physiological sensors) that can be prohibitive. There are also webcam-based multimodal data collection toolkits, such as MediaPipe [17] which provides various computer vision algorithms (e.g., face detection, hand/pose tracking, object detection, etc.) in one easy to use package for programmers. However, mediaPipe was designed to be used in various programming languages (C++, Python, JavaScript), for example for developing augmented reality or robotic applications, and not for multimodal data collection.

With these considerations in mind, we developed a new web-based multimodal data collection platform: the EZ-MMLA toolkit. This toolkit does not require physical sensors, such as dedicated eye-trackers or motion sensors, or any extra software to be installed locally. It runs in popular browsers (e.g., Chrome, Firefox, Safari) and is available to anyone for free around the globe at mmla.gse.harvard.edu. The EZ-MMLA toolkit is a significant step toward lowering access to multimodal data for researchers and practitioners. In the next section, we compare traditional ways of collecting multimodal data with the EZ-MMLA toolkit.

## 3. Multimodal Data Collection

This section describes the conventional approach towards multimodal data collection in research (Figure 1) and contrasts that with the new, proposed approach.

### 3.1. Conventional Multimodal Data Collection

Multimodal data collection has conventionally required setups involving physical sensors, proprietary tools, and external contractors. Data collection hardware can require complex setups and operating procedures that researchers are unable or unwilling to learn. Proprietary software or working with contractors can be expensive and unsustainable. Each data modality used in research comes with its own set of costs and challenges. Combining multiple data types from different hardware and software for a truly multimodal study requires additional coordination efforts and data analysis expertise. In sum, multimodal data collection has often been subject to the following constraints:Accessibility: Logistically challenging data collection and data analyses processes can restrict access to users without a strong technical background (c.f., computer scientists, engineers, statisticians).Affordability: While sensors are becoming more affordable as a whole, research-grade equipment still tends to be cost inhibitive. This results in multimodal data research being restricted to well-funded teams or to more commercially lucrative research areas.Invasiveness: Sensors created for research can still be bulky and invasive for human participants, impacting the state of the participant, their reaction to stimuli, or how aware they are of being observed. This can create noise in the data, or be the cause of Hawthorne effects and similar biases.

Issues of accessibility, affordability, and invasiveness have constrained the widespread adoption and scale of multimodal research. The complexity of data collection processes also make it difficult to develop a system of frequent assessment and redesign, which is integral in application studies.

Yet, with advances in computer vision, machine learning, and computer hardware, there are new possibilities for streamlining and improving the accessibility of multimodal data collection. Researchers can now replace a number of existing data collection methods with inexpensive and user-friendly software-based approaches [18]. For instance, heart rate tracking is traditionally performed with an electrocardiogram attached to sensors, placed on participants to measure electrical signals in the heart. Heart rate data can now be measured with a simple webcam and a computer vision algorithm that calculates a participant’s microscopic skin tone differentials to infer their heart rate [19]. While this measure may not be as accurate as electrocardiograms, it provides a reasonable proxy for heart rate measures while being minimally invasive and requiring no specialized hardware. We expect such web-based algorithms to significantly improve in their accuracy over the next decade with advances in computer vision and machine learning methods.

Established machine learning algorithms can now be feasibly used to develop alternatives to conventional multimodal data collection methods. With time, researchers may have to deal less and less with the restrictions and limitations associated with conventional methods for multimodal data collection. There is now the opportunity for multimodal research tools to be democratized [11]. Open-source implementations and libraries allow easier entry into the field, and there is also the possibility of tools, new methods, and particular pipelines being updated by a wider community of researchers with varying expertise.

### 3.2. Web-Based Multimodal Data Collection

Large machine learning applications can now operate online in real-time with support from open-source development teams such as Google’s TensorFlow.js. As such, our proposed approach for multimodal data collection is to utilize web-based platforms and applications for data collection. This platform is primarily intended to be used by researchers for capturing multimodal data during web-based experiments but can also be used by end users such as teachers and students. For instance, the tool has been used with consent to collect data from students in an introductory data analysis course to gain insights for teaching and to also serve as raw data for data analysis assignments.

For research, the web-based platform can serve as an online portal that researchers use to design and carry out multimodal research experiments. Researchers can set up simple experiments on the platform, or they can use other computer setups and have the platform running in the background for data collection. Participants will enter the platform on their end and engage with the experiment from either a specified computer or their own devices while carrying out the tasks of the experiment. During this time, the platform will capture and real-time process the video and audio of the participant through the computer’s webcam and microphone with user permission. The result is a processed, organized multimodal datasets that are ready for analysis. Ideally, all processing occurs real-time on the local computer so that no video or photos are saved. Figure 2 is a visual representation comparing a standard multimodal data collection process (left) with the proposed web-based solution (right).

The remaining sections of this paper describe our implementation of a web-based multimodal data collection system, the EZ-MMLA Toolkit. The toolkit is built using open-source implementations to collect multimodal data from video and audio input that can either be uploaded or captured real-time.

## 4. System Description

The EZ-MMLA Toolkit is a website that provides users with tools to collect multimodal data (Figure 3). This data can be collected from video and audio input, captured either in real-time or uploaded for post-processing. Some of the types of data that can be collected on the website are body posture, facial orientation, gaze location, emotion and heart rate.

The EZ-MMLA Toolkit utilizes JavaScript machine learning algorithms that run entirely within the web browser to capture data from video and audio input. These models offer a host of benefits compared to traditional models that are deployed on remote servers and accessed via API calls. A major advantage is that the computation needed to train the models and run predictions is offloaded to the user’s own device, rather than an external server. Therefore, the need to maintain an expensive remote machine is eliminated, which allows the EZ-MMLA Toolkit to have stable performance even with high user traffic. Furthermore, many of our models run on Tensorflow.js, a machine learning library in JavaScript, which is able to automatically determine a device’s unique set of constraints, like available WebGL APIs, and handle the model configurations. This makes our application compatible with various devices. It also allows for seamless integration of the models into our application, as we do not need to manually code for device constraints.

Additionally, direct browser integration of the tools makes it easy to connect models to device inputs like microphones and webcams. It also allows for personal video and audio data to be processed within the user’s own devices; so users do not have to worry about the security risks of sending their sensitive information over a network. Finally, the models allow for rapid real-time inferencing on the client-side, leading to more comprehensive data generation and smoother user experiences. The EZ-MMLA Toolkit is free to use and publicly available; it can be used by all educational practitioners, researchers, and learners.

The EZ-MMLA Toolkit website has been designed with a user-friendly interface that is minimalist and does not require any pre-requisite technical knowledge to use. On the tools page, users can select from an index of data-collection tools, provided in both a list and a grid format, which correspond to a multimodal data type. Figure 4 illustrates the grid-format of this page. When a user selects to run a tool, they are directed to the respective tool’s run page where they can either execute data collection in real-time through a webcam or microphone feed or upload a recording. Figure 4a is a screenshot of the pose-detection tool from the EZ-MMLA Toolkit. Once the tracking is complete, the user is shown a short summary of the data that was collected, and then either prompted to download the data as a CSV file (Figure 4b) or restart the tool. Details of the structure of the CSV file and the machine learning models underlying each tool can be found under the tool’s “Learn More” page. These tools can be run in a browser in the background while students complete classroom work on their devices, and this data can then be gathered by the instructor and analyzed to determine student learning.

While we are regularly adding new data collection channels, the EZ-MMLA toolkit currently provides the following data collection capabilities: eye-tracking, pose detection, hand tracking, emotion detection from facial expressions, emotion detection from speech, heart rate detection, object detection and lower-level detectors (e.g., color tracking and fiducial tracking). Figure 5 contains screenshots of six tools offered by the toolkit:

Finally, the EZ-MMLA toolkit facilitates multimodal data fusion. Typically, data fusion is a difficult problem to solve because different devices (e.g., eye-tracking goggles, physiological wearables, motion sensors) have their own data processing units. These units often have their internal clock, collect data at different frequencies and sometimes cannot guarantee a consistent data collection rate. In comparison, when users work with a single video recording, the video can be post-processed by a variety of tools on the EZ-MMLA website, for example to extract participants’ body posture, hand gestures, heart rate and emotion from facial expressions. Each row of the resulting datasets is tagged with the corresponding frame number of the video, therefore the datasets can be easily joined on this column. While the EZ-MMLA toolkit does not combine data for users, this can easily be achieved using a programming language (e.g., in 1–2 lines of code using the library Pandas in Python), a third-party software (e.g., Tableau, Rapidminer) or even manually (e.g., by copying/pasting additional columns using a spreadsheet manager such as Microsoft Excel). Using a single video as a data source makes multimodal data fusion straightforward—and avoids the need for complex solutions typically used for this type of problem (e.g., time synchronization with an external server, or direct clock synchronization across devices). In future iterations of the website, we are planning to include dedicated tools for multimodal data extraction and fusion (i.e., where one video can be processed to generate one unique csv file with multiple modalities).

### 4.1. Comparing the EZ-MMLA Toolkit with Traditional Data Collection Tools

While computer-vision algorithms provide users with new ways of capturing multimodal data, they are not interchangeable with physical sensors. Sometimes webcam-based tools are equivalent to (or better than) dedicated sensors, and sometimes dedicated sensors provide better data. Table 1 provides a summary of the differences between these two approaches for a few modalities:

In terms of accuracy, we can see that webcam-based data collection tools tend to be less accurate than dedicated sensors, e.g., Tobii 4C/5 is more accurate than Webgazer for eye-tracking and Empatica E4 is more accurate than Video Magnification for heart rate detection. However, the webcam-based sensor algorithms have been steadily improving as consumer hardware becomes faster, higher resolution, and more affordable. For example, the Microsoft Kinect sensor was the cutting-edge skeletal motion tracking technology for a long time. With the advent of OpenPose, PoseNet, and other related computer vision-based body tracking algorithms, regular cameras can provide better tracking accuracy (at least for 2D pose detection). These algorithms are very promising methods for performing data collection on a large scale, and some can achieve near real-time performance. We expect that many other algorithms will also significantly improve their accuracy over time.

Concerning cost, we can see from Table 1 that many physical sensors are more expensive (e.g., the Empatica E4 costs more than a thousand dollars) than their web-based counterparts. This limits the use of these physical sensors to more well-funded research teams or more commercially lucrative research areas. In contrast, many of the webcam-based models are open-source and the computation needed to run them is offloaded to the user’s device. This eliminates the need to maintain an expensive remote server to run the models, thereby allowing our application to be free to use and publicly available.

Moreover, webcam-based data collection tools provide many more benefits that physical sensors traditionally do not offer. While physical sensors can generally be unintuitive as they require domain expertise, proprietary tools, and external contractors to understand and use, our toolkit has been specifically designed with a user-friendly interface that anyone can use to collect data despite their technical abilities. Furthermore, a challenge of physical sensors is that they can be intrusive and unnatural to wear for participants (e.g., eye tracking headsets), which can create noise in the data. Therefore, it is preferable to use audiovisual modalities to interfere as little as possible while keeping real-time feedback, which the algorithms in our toolkit provide. Finally, data fusion can be a difficult task to solve with physical devices that typically have their own data processing units and internal clocks. The EZ-MMLA toolkit, on the other hand, tags each row of the resulting datasets with the corresponding frame number of the media, so different datasets can be merged on this column.

### 4.2. Possible Applications of the EZ-MMLA Toolkit

In this section we describe potential applications of the EZ-MMLA website, to provide examples of the potential of this toolkit. We categorize a few potential uses based on different audiences (researchers, teachers, learners, engineers).

First, social scientists can use multimodal data collection tools when conducting controlled experiments. As mentioned above, the website allows for real-time detection of body postures, hand movements, emotion detection, gaze direction, heart rate estimation, to name a few. Generating fine-grained datasets on these behaviors can speed up research by minimizing the need to hand-code video recordings. It can also provide a new lens for investigating social behaviors, especially when these behaviors are difficult to observe (e.g., subtle hand movements, physiological changes). A final advantage of the EZ-MMLA website is that it can also post-process video recordings; which means that researchers can generate new datasets on old studies to investigate new research questions. For example, a group of social scientists might be interested in questions of proxemics [12]; using PoseNet, they can precisely compute distances between participants and conduct fine-grained analyses of previously run experiments. Without PoseNet, this would require hundreds of hours of painfully annotating video frames.

Second, EZ-MMLA website can be used by learners and promote data literacy. Scientists can learn a great deal by collecting large multimodal datasets, so why not students? Given that the website is intuitive and easy to use, it can be used by young adults to explore data visualization, data analysis and data mining techniques. One promising application of this approach is through quantified self-projects [20], where users collect data on themselves with the goal of improving physical, mental, and emotional performance. Quantified self makes sense from a pedagogical perspective: instead of working on fabricated or foreign datasets (which is the most common approach in data sciences courses), learners work on datasets that they know intimately [11]. They are less likely to get lost in a sea of numbers, or to conduct appropriate data analysis.

Third, multimodal data collection can be used by teachers to monitor students’ learning. We also provide the source code of these tools, therefore, they could potentially be integrated in Learning Management Systems (LMS) and Massive Online Open Courses (MOOC). Traditionally, these data sources only capture the learner’s interactions with the learning system as simple log files [21]. This could be augmented with multimodal data, for example indicating students’ attention (from eye-tracking data) or emotional reaction (from facial expression) to various learning materials. There are of course data privacy and ethical issues that would need to be solved before this can be accomplished; however, these questions are outside the scope of this paper.

### 4.3. Case Study: A Course on Multimodal Learning Analytics

In the Fall 2020, the EZ-MMLA toolkit was used in a course on Multimodal Learning Analytics at a school of education in the Northeastern part of the USA. 52 students were admitted to the course (54% were female, 46% male). Researchers used the toolkit to collect multimodal data on students as they were learning the course material. Students used the website as part of an adapted version of an in-person course [11] that had transitioned online because of the COVID-19 pandemic. They signed a consent form to allow researchers’ use of their data, and completed various assignments involving eye-tracking, motion tracking, or emotion detection, using software such as Tableau and Rapidminder to visualize and analyze data. More than half of the students had little to no experience in programming and data visualization, and 81% had little to no experience with sensor data. In short, this population was ideal to test the EZ-MMLA website: participants were knowledgeable of social sciences methods but didn’t have any (or minimal) knowledge of working with multimodal sensor data. It provided us with feedback from novices (from a technical perspective), who at the same time were critical of the potential use of sensing technology in the social sciences.

The EZ-MMLA website was used in different ways in the context of this course: (1) students used it through weekly projects, where they collected multimodal data to build data literacy skills; (2) researchers collected gaze, pose and emotions data while students were watching weekly instructional videos on topics such as eye-tracking, emotion detection, pose tracking, machine learning or study design; (3) this data was used for a larger end-of-semester project, where students analyzed their own data collected during weekly videos in a “quantified self” manner; (4) finally, students conducted a final project that was completely open-ended: they had to choose a setting of their choice, formulate a research question, and answer it using multimodal data from the EZ-MMLA website. In the sections below, we describe some student feedback, final projects, and preliminary research findings.

#### 4.3.1. User Feedback

There are several sources of user feedback that were collected during the semester. One source is the anonymous weekly survey data from students, where three questions held potentially relevant comments for the toolkit. One question asked students to pick and describe an aspect of class they particularly enjoyed that week, the second an aspect of class that they felt needed improvement that week, and the third question prompted students to report their experiences using the EZ-MMLA toolkit. Additionally, we included a usability instrument in the survey [22]. Students were also prompted to give their feedback during class, sometimes orally and sometimes in a chat room. Lastly, we asked students to share how they might use what they learned in the course after the semester.

A total of 504 open-ended response sets were collected over the course of 12 weeks, and 82 open-ended comments were identified as being relevant to the EZ-MMLA toolkit. We conducted thematic analysis (see [23], for an overview) on the combined data to identify key themes. Taking an inductive approach, preliminary codes were assigned to the data, which were then iteratively organized into two main themes (positive, negative) and seven sub-themes (interest in one’s own data, authenticity, accessibility, technical issues, steep learning curve, data quality, privacy) on how students found the experience (Table 2). The insights on usability and future usage are presented alongside the seven sub-themes.

*Usability*. To assess the usability of the website, we used Brooke’s “quick and dirty” System Usability Scale (SUS [22]) which comprises 10 questions that users rate on a 5 point scale from “Strongly Disagree” to “Strongly Agree”. Questions for example included “I found the system unnecessarily complex”, “I think that I would need the support of a technical person to be able to use this system” or “I would imagine that most people would learn to use this system very quickly” (half of the questions need to be reverse-coded). The final score is between 0 and 100, where [24] identified 7 levels (i.e., “worse imaginable”, “awful”, “poor”, “ok”, “good”, “excellent”, “best imaginable). 29 students completed the SUS instrument. One response was an outlier (i.e., beyond two standard deviations), and was removed from the data. The final score of the EZ-MMLA website is 71.94 (SD = 14.84), which can be considered “good”. The lowest scoring questions were “I think that I would like to use this website frequently” (mean = 3.5), which can reflect not a usability problem, but how relevant the website could be in their daily lives; and “I found the website unnecessarily complex” (mean = 3.6, reverse-coded), which suggests that the website could be simplified from a user’s standpoint.

*Future usage*. Students also gave open comments about what future uses the data analytics tools, including EZ-MMLA, might have for their work and studies. Several commented that they saw new use for machine learning in education, such as in assessment or tracking learner engagement, while others mentioned wanting to use EZ-MMLA for research in UX design, education and marketing. Encouragingly, a number of students remarked that they felt “intimidated” by machine learning and data analytics coming into the course, but realized it is more ”feasible and inexpensive” than they thought. Many explicitly mentioned that their lack of technical background coming into the experience did not hinder their ability to “build something meaningful in a short time.”

*Interest in one’s own data*. The most prominent positive theme observed throughout responses was how collecting and analyzing one’s own data (i.e., data collected on the student) made the process more interesting and intuitive. One student noted that working with her own data allowed her to “feel more invested in the final product and assignment”. Students expressed awe at how they were able to find meaning from their own data, through visualizations and analysis, with one student noting that the real-time visualization of her own eye movement “felt almost like telepathy.” Looking at the quantified indices of their behavior also gave students a chance to “reflect on my own experience”, hinting at potential self-reflection processes triggered by seeing data on their own studying behavior.

*Authenticity.* Students seemed to appreciate the fact that the data collected by the toolkit was, in the words of one student, “massive datasets that are generated by real MMLA tools.” Students were motivated by the fact that the data was authentic, and were also able to get their hands dirty in data analytics—one student noted that “the opportunity to work with data collection directly… helps me to understand [the] limitations of data.” Students were mixed in their perceptions of the ‘messiness’ of the data. One noted “I liked being able to practice cleaning and analyzing my own data, rather than having pre-cleaned data handed to me”, while others wished for “cleaner data.” More students, however, wanted to “learn more about [the] data cleaning process that should normally be done with these data”. This perceived authenticity presents an opportunity for the EZ-MMLA toolkit in data science education; it is well known that authentic problems that are directly related to students foster engagement, motivation and deep understanding [25,26].

*Accessibility.* Several students noted that the functionalities and data collected by the toolkit were intuitive to access and understand, despite their lack of prior expertise. One student stated that the activity using the emotion detection functionality of the toolkit “far exceeded my expectations in terms of what is possible to do as a student in our 3rd week of class”, and another felt the tool was “simple and straightforward to use”. Similarly, students used playful language (“play around with sensors”, “play around with data”, “experiment with all the different sensors”, etc.) to describe their interactions with the toolkit.

*Technical issues.* The most prominent negative theme that emerged, on the other hand, was the frustration caused by technical issues. The most common issue was the website simply being slow and laggy for some students. A few reported that their laptops were running slower than usual when running the toolkit, likely due to some laptops hardware being unable to support the computations for the toolkit and other applications at the same time. Another issue was reported from students located in Asia, where proxies limited their interaction with the website. These issues were resolved by asking students to close other applications on their laptop, and by using a VPN service to connect to the school’s network. Students in later weeks reported seeing improvements to how smoothly the website ran. The downloading function also caused some trouble, caused by modern browsers’ security features: only the current “active” tab can access webcam data, which means that data collection stops when another tab is selected. This issue was resolved by explaining this issue to students and asking to either keep the tab active or by using data collection from a different browser. A similar issue was found on students using Windows, where webcam data can only be accessed by one application at a time (on Unix-based operating systems, this is not the case). Lastly, some students reported having to tinker with the setup (e.g., change browser, clear cache) for the toolkit to work properly.

*Steep learning curve.* While some functionalities felt intuitive and accessible for some students, other functions felt difficult to grasp for others. Some students reported re-attempting data collection a few times before getting it right, and asked questions such as “How can we check if data is being correctly collected?” Some students explicitly asked for examples and additional documentation. For instance, on collecting data for skeletal tracking, one comment was “I wish we knew an average number of frames that should be collected”.

*Data quality.* A few comments pointed out the limitations of the data collected by the toolkit. One type of limitation was innate to the toolkit, such as students noticing that PoseNet had some difficulty capturing rapid movement. Another type of quality concern arose from the unique context of students being privy to the exact setup and inner workings of the subject when working with their own data. This included concerns about how they had been conscious of the sensors and modeled their behavior to get ‘good’ data, or how they were concerned about the data being inaccurate because they were using a dual monitor setup, or had repeated the same experiment a few times before collecting the final dataset.

*Privacy.* Specific to the data collected on the learning portal (i.e., when students were watching the instructional videos), a few comments expressed understandable unease at the thought of a camera continuously collecting data—even though researchers explained what kind of data was collected and participants were given the option to opt out. One student in particular noted that she felt nervous because she was unsure whether the website was collecting data or not at any given point. Another student responded in the early weeks that he did not know what the data collection website was doing, and if it meant that the camera was on, that felt privacy-invading. During the semester, this issue was addressed by adding a feature that allowed students to access the data collected on them while they were watching instructional videos. In the future, we hope to further mitigate this issue by adding a red/green light button to notify users on whether the camera is on or not, as well as by providing on-site data visualization functions to help users grasp the exact nature and scope of the data being collected (e.g., no image or video is saved or exported, only x, y coordinates of eye, nose, ear, hand, etc.).

In this section, we describe a few final projects that students conducted in the context of the multimodal learning analytics course (Figure 6). While students couldn’t run full experiments, they were told to look for trends and extrapolate potential results based on pilots (while recognizing the limitations of using a small sample size).

In the first project, students were interested in “augmenting” storytelling with custom digital accessories (e.g., helmets, shields) and backgrounds (e.g., mountains, fictional places). They designed a story that was told to two different audiences, and they captured participants’ emotions (from facial expressions) and heart rate during the story. Compared to a control group, they found that participants in the AR condition experienced a higher heart rate but there were no differences in facial expressions. They concluded that social norms might have influenced participants’ behavior, and that heart rate data might have captured differences that were not visible.

In the second project, students used recordings of the 2020 presidential debates to see if they could identify particular behaviors when the candidates were telling a lie. They ran PoseNet and emotion detection algorithms on the videos to generate multimodal datasets. They found that one candidate was found to be more “disgusted” and less “happy” when telling a lie, while the other candidate looked more “sad”. Additionally, the first candidate seemed to raise one shoulder (generally the right) and sometimes tuck his chin in when making false statements.

In the third and final projects, students investigated child-parent interactions during math lessons: using clustering analysis on pose data, they found two distinctive patterns of parental action (interactive vs. inactive). Exploratory analysis identified child gender, parental level of math anxiety, and parental math attitudes, as potential contributors to parents’ non-verbal interactive patterns with their child.

In all projects, students demonstrated a skilled use of the data collection tools while keeping in mind limitations of the technology (e.g., small sample size, potential noise in the data, alternative interpretation of the results). The most striking observation, however, was the breadth of projects. This suggests that the EZ-MMLA toolkit allowed them to answer research questions that would have been difficult to tackle otherwise.

#### 4.3.2. Preliminary Research Questions

Finally, the researchers are currently analyzing the multimodal data collected while students were watching the weekly instructional videos. As a reminder, we collected pose data using PoseNet (including the x, y location of each body joint), eye-tracking data (using webgazer [27]) and emotion from facial expressions. Over the entire semester, this generated 1,308,784 rows of eye-tracking data, 1,392,260 rows of emotion data, and 1,429,705 rows of pose data from 12 videos and 52 students (for a total of 4,130,749 rows).

We are currently investigating the following research questions using this dataset. For example, what are behaviors correlated with learning scores (as measured by the test administered after the video)? We are exploring simple metrics, such as the percentage of time that students were looking at the video (from the eye-tracking data), whether they were expressing positive or negative affects (from the emotion data), or if distracted behaviors, occurred (for example unexpected high velocity movements from the pose data). We are also exploring more sophisticated features, such as measures of synchrony with the instructor. For example, joint visual attention (JVA; [28]) has been shown to be correlated with learning in collaborative learning groups. Since we also collected eye-tracking data from the instructor, we can compute JVA measures between students and the instructor (i.e., do students look at the same place on the slides as the instructor?). Similarly, we can compute measures of affective synchrony (i.e., are learners/teachers expressing similar emotions at the same time?) or pose synchrony (i.e., are they adopting similar body postures and gestures)? Human mimicry has long been observed to occur when humans work jointly on the same task [29]. An hypothesis that could be tested is whether they are related to conceptual understanding during instructional videos.

In short, multimodal data can provide us with meaningful predictors for students’ learning. Not only can it describe and predict learning, but it can also potentially offer clues to understand why some students are obtaining lower scores on the learning test (for example if they were not paying attention to the “right” thing, i.e., what the instructor was looking at).

#### 4.3.3. Website Usage

Finally, we provide some data showing how the EZ-MMLA toolkit has been used since we started collecting analytics in early 2021 (the platform was also used for a Fall course in 2020, which is not included in the numbers below). Figure 7 provides some additional data in the number of daily visits (a), users’ origin (b) and tools used (c). We can see that the website has been consistently used since we started collecting analytics, with a peak of 150 users April/May 2021 (3600 unique users). Users viewed 19,163 pages, with the front page and the Tools page receiving most views, followed by PoseNet (1.3k views) GazeCloud (503 views). Users were 66 countries, primarily from the USA, Canada, Mexico, Brazil, central Europe, India and Australia.

## 5. Discussion

In this paper, we described the design of a multimodal data collection website, the EZ-MMLA Toolkit, which serves as a novel approach for generating data through video and audio streams directly in the browser. The application poses several advantages over conventional data collection methods, such as physical sensors and interaction data with learning platforms. Namely, the EZ-MMLA Toolkit makes data collection more accessible, unintrusive, low-cost and ethical, without compromising the quality of data generated. We further described potential applications of the toolkit and presented a case study where the website was used in a course on multimodal learning analytics. We reported feedback from students, their final projects, and preliminary research questions that can be answered by analyzing multimodal data from learners. We describe some limitations below and future steps for this project.

### 5.1. Limitations

Based on our findings, we observed three main set of limitations (usability, web-reliance and accuracy):(1)Usability: The use of web-based applications can still be challenging for some users—especially for certain learner types or when dealing with technical terms and instructions that might be difficult to understand. Additionally, the proposed approach has not been extensively tested across an adequately diverse pool of users. In particular, there is little testing on users with disabilities, where it is likely that we will find usability concerns.(2)Web-reliance: The proposed approach inevitably is limited by the accessibility of a stable internet connection—which is not possible for some user groups. Additionally, the reliance of our proposed approach on web platforms complicates the implementation of machine learning models because they will need to be re-optimized for use online. During development, we found that most of the most accessible and relevant machine learning libraries and software for multimodal data collection were implemented in languages such as Python and C++ and were not developed with web applications in mind. As such, this limits the range of usable machine learning libraries and poses a significant resource cost if models need to be reimplemented, reformatted, or translated for use in an online toolkit.(3)Accuracy: The accuracies of several implemented models leave to be desired (mostly because they are open-source projects) and limits the research potential of the toolkit. Furthermore, the accuracy of the data collected has not been extensively tested at the time of writing due to the novelty of our platform. This is also an important limitation because it restricts the overall usability of our tools as researchers will not have a strong understanding of the validity and integrity of the data collected.

### 5.2. Future Work

Considering the above limitations, we describe areas for future development below. First, there are additional computer vision tools that can be added, or better tools for particular data streams (e.g., emotion detection). We are also interested in providing variations of the same model, trained on different datasets to provide more accurate predictions for certain situations (e.g., emotion detection tailored to particular ethnic groups). This would also introduce the idea of bias in machine learning models, which would be useful for learners who are using the website for educational purposes. Second, and related to the first point, we are considering adding some tools and modules that make models more understandable to end users (i.e., understandable AI; [30]). These tools could provide information about the most informative features used when making predictions. Third, we would like to add a “data quality inspector”, so that users can better identify issues with the data. This could include noise or missing data detection. These tools are important both for research and educational applications, so that users can make better inferences from the data. Fourth, we are considering building “meta multimodal tools” that directly combine modalities together for specific situations. For example, one can imagine building a tool that captures dyadic collaboration and automatically generates relevant features for this kind of setting (e.g., joint visual attention, body synchrony, emotional mimicry, turn taking behaviors, etc.). Finally, we are also considering adding community building features, so that users can link projects that use the EZ-MMLA toolkit to the website. This would facilitate knowledge sharing, and inspire users to try analyses that others found to be successful.

## 6. Conclusions

To conclude, this paper makes the following contribution to the fields of multimodal data collection and learning analytics: (1) The paper proposes a web-based approach towards multimodal data collection and learning analytics and provides a comparison of this approach to existing conventional methods. (2) The paper provides an implementation of a web-based multimodal data collection website. (3) The paper evaluates the proposed approach through a case study: an integration of the web-based multimodal data collection website to a course taught to graduate students in education.

Overall, this paper finds that the results from preliminary examination and from the case study are encouraging. The prototypes have been broadly successful and indicative of how the proposed web-based approach might be viewed as a viable new medium for MMLA. Crucially, as highlighted by Schneider, Reilly, & Radu [11], our findings suggest that there is the potential to overturn the conventional MMLA pipeline and democratize multimodal research in the social sciences.

## Figures and Tables

**Figure 1 sensors-22-00568-f001:**
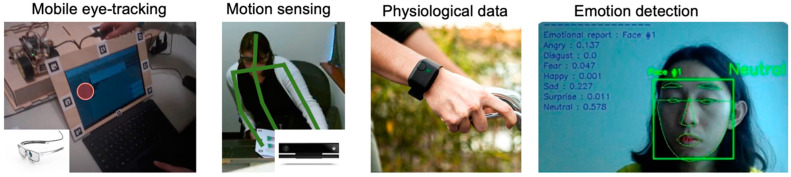
Use of sensors in a typical multimodal study. Left: mobile eye-tracker. Middle: Motion sensors and Electrodermal wristbands (Empatica E4). Right: emotion detection from facial expressions. Mobile eye-trackers are expensive wearable sensors and data processing software tend to be proprietary. The Kinect sensor is more affordable but does not come with an easy-to-use data collection package; recording the data requires some programming knowledge. The Empatica E4 is also relatively expensive and data extraction requires proprietary software.

**Figure 2 sensors-22-00568-f002:**
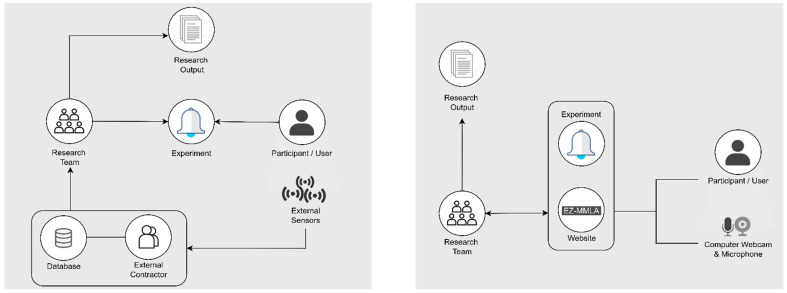
The image on the left represents a typical multimodal data collection process, involving physical sensors, proprietary software, or external contractors. The diagram on the right represents how data collection in the EZ-MMLA website is carried out locally with JavaScript models.

**Figure 3 sensors-22-00568-f003:**
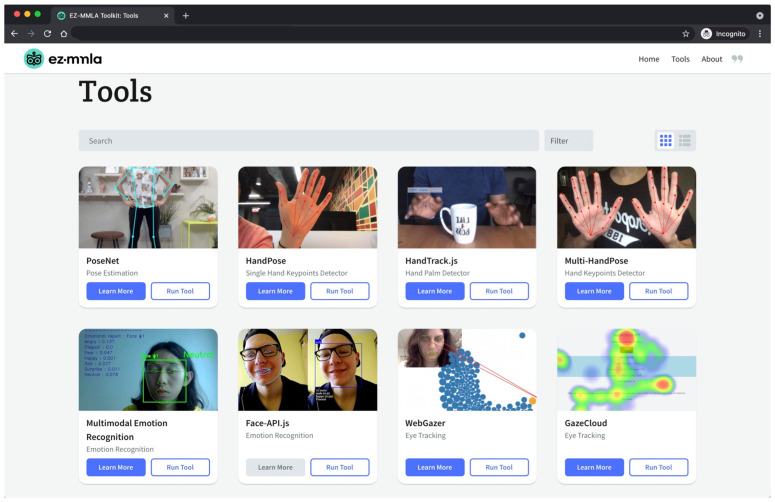
The Tools page of the EZ-MMLA toolkit. This page consists of a list of tools available for multimodal data collection. Each tool contains a link to start running the tool and a link to learn more about how the tool works. Figure 4 shows the page for running PoseNet, which opens up when the user clicks on the “run tool” button.

**Figure 4 sensors-22-00568-f004:**
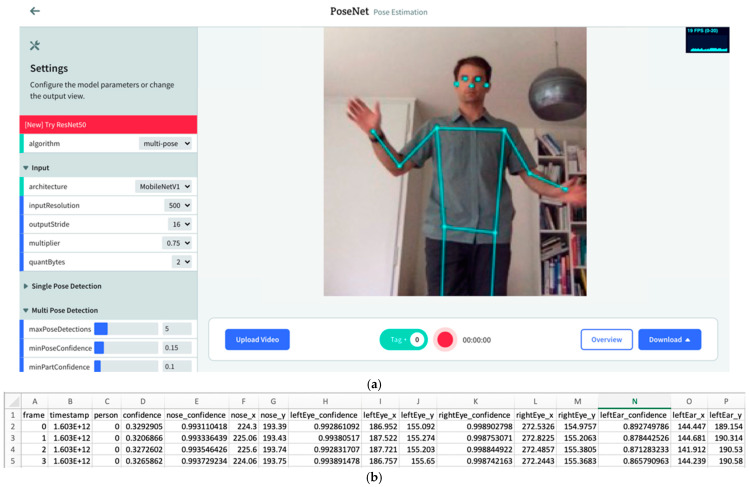
(**a**) The pose-detection tool’s data collection page from the EZ-MMLA Toolkit (PoseNet). This page displays a real-time video recording of the user, with the model’s predicted pose coordinates overlaid on top. In the top-right corner of the screen, there is an information box displaying the rate at which each frame is being processed per second by the model. In the left corner, there is a controller to change the various configurations of the model as well as the display output. At the bottom of the screen, there is a recording bar that contains (1) a recording button to start and stop data recording, (2) an overview button to provide a summary of the data collected in a recording, (3) a download button to save the generated data as a CSV file, (4) an upload video button for processing pre-recorded videos, and (5) a tag button to add tags to the data as the recording takes place. (**b**) A snippet of the CSV file of the data collected by PoseNet (i.e., frame number, timestamp, and for each body joint, the x, y predicted coordinates and the associated confidence level of that prediction).

**Figure 5 sensors-22-00568-f005:**
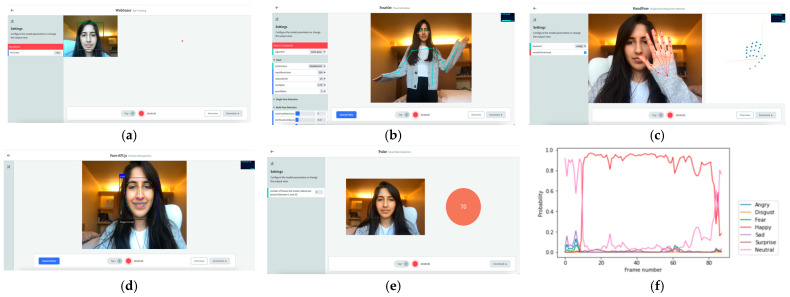
Screenshots of some of the tools featured on the EZ-MMLA website. (**a**) eye-tracking (using webgazer.cs.brown.edu, accessed on 10 October 2020); (**b**) pose detection (using ml5js.org/reference/api-PoseNet, accessed on 10 October 2020); (**c**) hand tracking (using storage.googleapis.com/tfjs-models/demos/handpose, accessed on 10 October 2020); (**d**) emotion detection (using github.com/justadudewhohacks/face-api.js, accessed on 10 October 2020); (**e**) heart rate detection (using photoplethysmography from github.com/avinashhsinghh/Heart-Rate-Detection, accessed on 10 October 2020); (**f**) a sample output of emotion detected from speech (using github.com/maelfabien/Multimodal-Emotion-Recognition, accessed on 10 October 2020). Besides these six, the website offers additional data collection tools.

**Figure 6 sensors-22-00568-f006:**
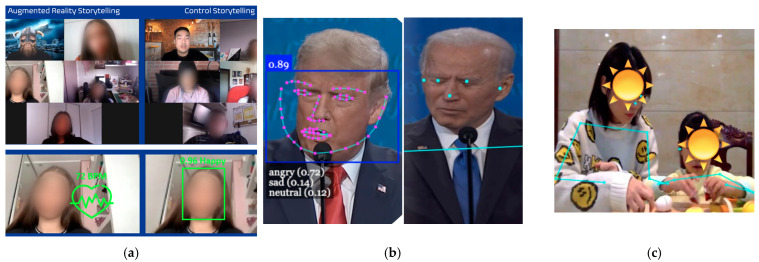
Examples of final projects (from left to right): (**a**) comparing the effect of augmented story-telling with normal storytelling on the audience’s heart rate and emotions; (**b**) analyzing facial expressions and body postures during presidential debates to compare moments where participants lied or told the truth; and (**c**) investing parents’ non-verbal interactive patterns while teaching math concepts to their children.

**Figure 7 sensors-22-00568-f007:**
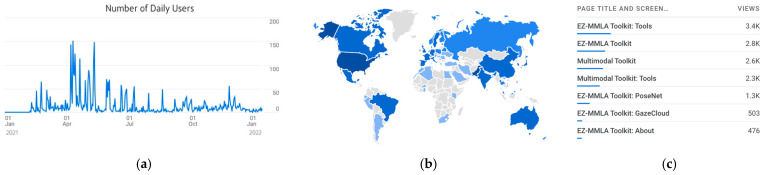
The mmla.gse.harvard.edu website receives frequent daily visits (**a**) from 66 countries all over the world (**b**). The main tools being used are PosetNet and GazeCloud (**c**).

**Table 1 sensors-22-00568-t001:** Comparison of Physical and Webcam-based sensors across different modalities. Exemplar tools are underlined. Pros and cons are described below each tool.

Modality	Physical Sensor	Webcam-Based Sensor
Skeletal tracking	Kinect v2 © Pro: 3D tracking Con: Noisy in case of occlusions; discontinued	PoseNet Con: 2D tracking (no depth) Pro: more accurate in case of occlusions
Eye-tracking	Tobii 4C/5 © Pro: accurate Con: cost	Webgazer Pro: can be integrated to any website Con: less accurate
Hand tracking	Leap Motion © Pro: easily integrated to any headset	Google AI Hand Tracking Pro: as accurate as physical sensors Con: slow for multiple hands
Physiological sensing	Empatica E4 © Pro: electrodermal (EDA) data Pro: accurate heart rate (HR) data Con: cost	Video Magnification [19] Con: no EDA data Con: noisy HR data
Applicable to all modalities	Con: cost (requires a dedicated device/license) or the use of an SDK to access the data Pro: data quality is guaranteed	Pro: easy to set up, anyone can collect data Pro: data is processed locally (via JavaScript) Con: data quality can vary

**Table 2 sensors-22-00568-t002:** The main themes identified from 504 response sets that were collected over the course of 12 weeks.

Theme	Examples
Interest in one’s own data	“I like VERY MUCH using data generated by ourselves” “I like learning and analyzing my own learning process”
Authenticity	“a real world scenario” “I liked that it was grounded in something real-world and relevant to the class”
Accessibility	“Easy to use, time limit had to be monitored. Great resource.” “The fact that we got to actually use the Emotion Detecting tool and analyze it this week far exceeded my expectations in terms of what is possible to do as a student in our 3rd week of class!”
Technical issues	“The eye tracking data collection website tend to make my laptop run slowly.” “[The videos on the data collection website] take forever to load and keep buffering.”
Learning curve	“I wished there is a tutorial to the data collection website, there are a lot of functions I knew after I finished collecting my data.” “When I was collecting eye-gaze data from gazecloud, it took me several tries to figure out how I can both read and let gazecloud track my eyes.”
Data quality	“I wish I was made more cognizant of how [the] quality [of] the data being collected is as I was being recorded (because I forgot a lot of the time). If there was a camera to show us how the camera is detecting our position, I might change my behaviors so that I have my webcam screen on/not be leaning back out of view” “Using the gaze recorder on Chrome, I had it in another desktop view and it did not record my gaze while I was reading my classmate’s slide. So I had to do it over again, and tried to replicate where my attention was but, clearly, it is already manipulated data.”
Privacy	“my camera was on and I had no idea if it was recording or not.” “I’m not sure what is the data collection website, if that refers to having camera on while watching the asynchronous video, it felt weird and I feel my privacy is being violated a little.”

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
