# Peer review of "Augmenting Social Science Research with Multimodal Data Collection: The EZ-MMLA Toolkit"

_sensors, 2022, doi:10.3390/s22020568_

Round 1

Reviewer 1 Report

Authors present an aggregation of some open source projects whose aim are to use household equipment (e.g. web camera and microphone) to sense some human aspects, such as heart hate (video augmentation/magnification), pose tracking and detection (mainly statistical and ANN) including gaze, and mood detection (mainly ANN).

Although the proposed toolset has some other variables that could be sensed, I could not find any new proposal. I must recognize the work done on porting all the tools to one place and to one AI framework in order to allow browser execution without any installation. This work is superb and I did use the heart rate monitoring while reading your paper. 

The novelty of the work is on applying the toolset to social sciences, specifically to education effectiveness evaluation. However, authors does not explore the study nor present relevant data on the toolset application, resuming to a report of application. 

I've also read the ACM paper (LAK21) and could not identify the extension proposed on this version.

Because of the exposed above, I would like to hear from you on the next revision round:

1) How this work fits into Sensors aims and scope (please consider the novelty above)?

2) How the extension improves the previous publication on ACM?

3) Can you present data and data analysis about either your application (on the students or even generated by one of them)?

4) I understand how multimodal sensor fusion would work in your scenario, but I could not find any clue on how your toolset does this. It seems a data collection framework on what I could understand. Could you please clarify this and also present some examples in your rebuttal letter?

Thank, hope to hear form you soon!

English are far beyond I can evaluate as a non native speaker, so I just found a few typos. I had no problems with the text itself.

Typos and others:

L39: double point (..)

L222: Figure can be improved. I suggest to use a higher definition figure in a more horizontal fashion to benefit the page width.

L240: bogus space before endpoint.

Reviewer 2 Report

The paper presents the EZ-MMLA Toolkit, i.e., a website designed, developed and demonstrated for multimodal data collection and learning analytics.

The paper is well organized and fluently readable. Its motivations and ideas are convincing. Also, its case study demonstration is interesting. Having said that, there are some issues, which are discussed below.

To begin with, despite its acknowledged technical and applicative merits, the scientific novelty behind a Web site appears not to be high.

All contributions of the paper have to be explicitly summarized in the Introduction along with their novelty with respect to previous research efforts in the literature.

A plan of the paper has to be added to the Introduction.

A more extensive and comparative discussion on the advantages and disadvantages of the authors’ contribution is required.

Finally, in the absence of a quantitative evaluation, the demonstration of the potential of the authors’ contribution through an additional case study would strengthen its qualitative assessment.

Reviewer 3 Report

This is a very interesting paper, well presented

and offers a great contribution to multimodal data collection

Author Response

Thank you!

Round 2

Reviewer 2 Report

The authors’ review improved and strengthened the original paper, which is now suitable for publication as long as the clarification and justification of novelty provided by Reponse 1 is fully reported in the article.